# Consumer Acceptance and Market Potential of Iodine-Biofortified Fruit and Vegetables in Germany

**DOI:** 10.3390/nu13124198

**Published:** 2021-11-23

**Authors:** Ann-Kristin Welk, Ruth Kleine-Kalmer, Diemo Daum, Ulrich Enneking

**Affiliations:** Faculty of Agricultural Sciences and Landscape Architecture, Osnabrück University of Applied Sciences, Am Krümpel 31, 49090 Osnabrück, Germany; r.kleine-kalmer@hs-osnabrueck.de (R.K.-K.); d.daum@hs-osnabrueck.de (D.D.); u.enneking@hs-osnabrueck.de (U.E.)

**Keywords:** dietary supplements, functional fresh food, health claims, iodine biofortification, mineral micronutrients, nutritional claims, target groups

## Abstract

Biofortification of food crops with iodine is a novel approach to preventing iodine deficiency in humans. The present study analyses the consumer target groups and the market potential of iodine-biofortified fruit and vegetables in Germany. For this purpose, an online survey of 1016 German fruit and vegetable consumers was conducted to investigate the acceptance of different product categories as well as relevant criteria for the market launch. The results show that iodine-biofortified fruit and vegetables are particularly attractive to consumers who purchase at farmers’ markets, organic food shops, and farm stores. Out of this group, 39% of consumers rate such iodine-rich foods as very appealing. They attach importance to food that naturally contains iodine and prefer produce from integrated domestic cultivation. With their focus on sustainability and naturalness, this group of consumers clearly differs from typical users of dietary supplements, who are primarily concerned with health benefits. However, overall about 85% of respondents would prefer biofortified fruits and vegetables to supplements to improve their iodine supply. The greatest market potential for iodine-biofortified fruit and vegetables is to be expected in supermarkets, as this is the preferred food shopping location for most consumers. A total of 28% of those who buy here rate the biofortified foods presented as very appealing. Nevertheless, a successful market launch requires that the benefits of the new products are communicated according to the potential consumer group needs.

## 1. Introduction

Iodine is an essential trace element for the human body. Among others, it contributes to normal thyroid function and formation of thyroid hormones, normal cognitive abilities, energy metabolism, and function of the nervous system [1,2]. For maintenance of these functions, adults and adolescents require a daily iodine intake of 150 µg [3]. According to World Health Organization (WHO) estimates, around 1.9 billion people worldwide are insufficiently supplied with iodine [4]. This corresponds to about 28% of the world’s population. In Europe, the issue of iodine deficiency is even more widespread, affecting 44% of the population on average [5]. 

Studies conducted in Germany show that 32% of adults and 44% of children and adolescents fall short of the estimated average iodine requirement [6,7,8]. Overall, the iodine status in the German population has deteriorated in recent years to the point that this country is now once again classified as an iodine-deficiency area according to the WHO criteria [9]. Various reasons have probably contributed to this development. Sales of iodized table salt in German food retail fell from 32,328 tons in 2015 to 29,273 tons in 2019 [10]. However, only 10% of salt intake is due to household consumption. The majority of salt (77%) is consumed through processed foods, such as bread, meat, and dairy products, which are mainly produced with non-iodized salt [11]. Therefore, iodized salt contributes only 28% of the total salt intake [12,13,14]. As a preventive measure to avoid cardiovascular diseases, a general salt-reduction strategy is pursued in Germany [15,16]. In a recent consumer study, about a quarter of the surveyed participants stated that they already attached importance to a low-salt diet or wanted to reduce their salt consumption in future. The same study also revealed that only about half of the population is sufficiently informed about iodine and its requirements for the body [12]. This is also reflected in increased demand for sea salt or Himalayan salt, which are considered to be of particularly natural or premium quality but are not enriched with iodine [12]. In addition, a trend towards plant-based nutrition can be observed, consisting of diets that may be associated with inadequate iodine intake [17]. Currently, 9.2% of Germans from 14 years of age classify themselves as vegetarians and 1.6% as vegans [18]. Food crops, such as fruit, vegetables, and cereals, usually contain less than 1.0 µg of iodine per 100 g of fresh mass. This is due to the fact that soils are low in phytoavailable iodine, and therefore, the absorption of this trace element by plants is quite limited [19,20].

Biofortification is a new method to improve iodine supply that is independent of salt consumption and suitable for a plant-based diet. In this approach, the iodine content is already increased in food plants during cultivation. In principle, classical breeding, genetic engineering, and agronomic techniques can be used for this purpose [21,22,23]. The latter method relies on iodine-containing fertilizers, which are applied to the soil or as foliar sprays onto the aboveground plant parts. Agronomic iodine biofortification has already been successfully tested on fruit, vegetables, and culinary herbs, such as apples, pears, tomatoes, lettuce, and basil [24,25,26,27]. From a consumption perspective, fruit and vegetables also appear to be suitable for iodine biofortification, as 71% of the German population eats them daily [28]. Furthermore, an earlier study on selenium- and iodine-biofortified apples at the Osnabrück University of Applied Sciences showed that about two-thirds of Germans are open to this form of enrichment [29]. Iodine-biofortified products are not yet available in the German food trade. However, in Italy, they have already been introduced to the market and are increasingly in demand [30].

If fruit and vegetables reach an increased iodine content of 22.5 µg/100 g fresh weight as a result of biofortification, health claims can be used in marketing [1,31]. Foods that beneficially influence one or more target functions of the body are classified as functional foods. This can be an improved state of health and well-being or a reduced risk of disease. Functional food is consumed as part of a normal diet [32]. Most often, these are processed products that are enriched with certain ingredients during manufacturing [33]. To distinguish from such common functional foods, the term “functional fresh food” is introduced in this paper for unprocessed, biofortified fruit and vegetables. Currently, around 42% of the population in Germany regularly consume functional food products [34]. The largest target group is represented by female persons over 45 years of age with high purchasing power [35]. However, young people between 20–39 years with a high level of education are also interested in functional food [36]. In general, it can be stated that the consumer group of functional foods depends on the type of product group. For example, cholesterol-lowering margarine is primarily consumed by a consumer group aged 50 or older, while soft drinks enriched with vitamins or minerals are preferred by consumers aged 19–34. A generalization of consumer characteristics is therefore only possible to a limited extent [33]. Consumers cite strengthening the immune system (56%), increasing energy levels (45%), and promoting brain and memory function (43%) as reasons for buying these functional products [35].

In addition to functional foods, dietary supplements provide a means for additional intake of iodine. They differ from functional foods in their dosage form. Unlike normal foods, dietary supplements are offered as tablets, capsules, or powders that contain ingredients, such as minerals and vitamins, in concentrated form [37]. The share of users of dietary supplements in Germany increased from 16% to 28% from 2013 to 2020. Women make use of them more frequently than men. Consumption also increases with age. In the age group 65 and older, about every second woman or every third man uses supplements in their diet [38,39,40,41]. The growing popularity of vegetarian and vegan diets among younger consumers (16–39 years) is accompanied by rising demand for dietary supplements in this age group as well [40]. Reasons for consuming dietary supplements include the desire to be supplied with all necessary nutrients and general health maintenance. The consumption of dietary supplements usually has no specific trigger. However, it can be observed that people who are active in sports are increasingly turning to supplements [40,42,43].

In contrast to dietary supplements and functional foods in general, potential consumer target groups for iodine-enriched functional fresh foods in Germany have not yet been analyzed. Therefore, the aim of this study was to examine the acceptance of German consumers towards iodine-biofortified fruit and vegetables, to characterize the target groups, and to estimate the market potential in food retailing. In view of this, the consumer survey that was conducted addressed the following research questions:What is the level of knowledge of the German population regarding the trace element iodine in terms of deficiency symptoms, and which food sources are preferred?Who are the potential consumer target groups of iodine-biofortified fruit and vegetables? What are their motives for buying these products?Do the target groups of iodine-biofortified fruit and vegetables differ from users of dietary supplements in their characteristics and motives?How should the market launch of iodine-biofortified fruit and vegetables be prepared conceptually? Which distribution channels in food retailing offer the greatest market potential for these functional fresh foods?

## 2. Materials and Methods

### 2.1. Data Collection

Data were collected in a quantitative anonymous online survey, conducted with German consumers in January 2020. For this purpose, a fully structured questionnaire was developed to collect a wide range of consumer insights. It should be emphasized that this survey is considered as an initial step in the preparation of a market entry for biofortified food, to be followed by others. Discrepancies may occur between consumer statements in surveys and actual shopping behavior, as shown for example in studies on ethnic consumer behavior [44]. Therefore, subsequent studies are planned to evaluate consumer acceptance of biofortified vegetables and fruits in real market tests.

The sample of the present quantitative survey (Table 1) is representative for age, gender, and place of residence of Germans. Furthermore, an exclusion procedure was used to ensure that the survey participants are all fruit and vegetables consumers and are those responsible for food purchases.

Respondents were recruited in January 2020 by respondi AG, a licensed provider of online household panels for consumer research. Based on our concept, they created the questionnaire using the Tivian software from Tivian XI GmbH based in Köln, Germany. Data-quality checks were carried out during the survey regarding the exclusion of incomplete interviews and interviews with a very short interview length. Furthermore, interviews that did not pass the question on the level of attention were excluded from the sample (Appendix B; Q22). A total of 1016 survey records were included in the statistical analysis.

### 2.2. Study Design

In order to answer the research questions mentioned in the introduction, the questionnaire was designed and divided into five sections. The questionnaire was based on our previous studies on selenium- and iodine-biofortified apples [30]. In parts, adjustments were made based on previous experiences and findings. The first section of the questionnaire after the screening questions was dedicated to purchasing habits and consisted of two questions: first, the place of purchase and, secondly, respondents perspective of “natural foods”. The place of purchase fulfilled the function of providing information about the subsequent distribution channel of the functional fresh food products. Identifying associations with the term “natural food” was supposed to provide information on which characteristics should be taken into account in the development and communication of iodine-biofortified foods. In the second part, the respondents were asked about functional food affinity and general knowledge about mineral micronutrients. The focus was on the effect and benefit of iodine and the appearance of iodine-deficiency diseases. Furthermore, the respondents were asked about their personal iodine sources in their daily diet and forms of fortification. This questionnaire section served on the one hand to validate the consumers’ level of knowledge about iodine and on the other hand to examine the importance of health claims when purchasing functional fresh food. In addition, this section provides information on the popularity of iodine-biofortified produce in direct comparison to salt, dietary supplements, and processed functional foods. It therefore shows a first general acceptance of iodine-enriched fruit and vegetables in comparison to other product categories. A concept test was the third and central part of the survey. For the concept test, the test persons were divided into two groups. Group A received a short information text about the current iodine supply in Germany and the consequences of iodine deficiency as well as the advice that iodine-enriched fruit and vegetables can contribute to a better iodine supply. The credibility of the information was underlined by the logos of the Federal Ministry of Nutrition and the evaluation “German Health Interview and Examination Survey of Adults” (DEGS) of the Robert Koch Institute (Appendix B). In addition, group A was asked about the health benefits of six health claims that can be used to highlight iodine-containing foods. Group B received no information on the added value of iodine-enriched foods. Furthermore, both groups were asked identical questions about the acceptance of biofortified apples, lettuce, tomatoes, and basil. To make it more concrete and easily imaginable for the survey participants to answer the iodine-biofortified fruit and vegetables acceptance test questions, one product was selected as an example for each of the four food groups: fruit, fruit vegetables, leafy vegetables, and potted herbs. These were apple, tomato, lettuce, and basil, respectively. The selection of the above-mentioned foods was based on two criteria: first, the suitability of the products for biofortification and, second, the popularity of the products in Germany. During the survey, each participant was first presented with all four iodine-biofortified products so that the product groups were first compared with each other in the acceptance test. In the next stage, the respondents were asked to choose one of the biofortified products. Subsequently, they were asked about the willingness to pay for the selected product and the assessment of properties in comparison to dietary supplements with iodine. The concept test was supplemented by five questions, including willingness to pay, reasons for purchase, and forms of biofortification, which were answered by all respondents. To keep the complexity of the section low, each participant was asked to choose a product (apple, lettuce, tomato, or basil) that they personally find most appealing before they were given the five questions. The aim of the test, including the random division of the test persons into two subgroups, was to investigate the influence of information on the acceptance and willingness to pay for iodine-biofortified fruit and vegetables as well as to identify preferred product characteristics (including health claim, labelling, and approach of biofortification) and analyze purchase motives of the consumers. After the concept test, a fourth questionnaire section on health aspects of food consumption was added. In this section, the participants were asked about general health statements as well as about the consumption of dietary supplements and special diets. The questions were intended to identify potential target groups of fresh iodine-biofortified fruit and vegetables, including their purchase motives, and to compare them with the target group for dietary supplements. Furthermore, the direct comparison of functional fresh food and dietary supplements gives an insight into the added values of the fresh products from the consumers’ point of view. The survey was completed with the collection of socio-demographic data, such as school-leaving qualifications, occupation, household size, and number of children in the household. These data were used to complete the target group allocation.

### 2.3. Statistical Analysis

By means of analysis of variance (one-way ANOVA), the mean values of groups A and B were compared in the concept test to determine whether additional information leads to a higher acceptance of iodine-biofortified fruit and vegetables among consumers. 

Regression analysis was used to identify target groups of iodine-biofortified fruit and vegetables and compare them with consumers of dietary supplements. For this purpose, the questions on concept acceptance, for groups A and B, were combined together, and a mean value was collected across all four food product groups (Appendix B; Q18, Q19). In this way, a 17-item scale was generated and used as the dependent variable in a linear regression model. Demographic (e.g., age and gender) and socio-demographic characteristics (e.g., education level and household size) as well as psychographic characteristics (e.g., dietary style and reasons for purchase) served as independent variables. In addition, the variables on labelling and health claims were included. In order to survey the target group of dietary supplements, an ordinal regression analysis was conducted with the consumption frequency of dietary supplements in three levels (regular consumption, occasional consumption, no consumption) (dependent variable) with the independent variables listed above. The selection of criteria was justified by the hypothesis that consumers of iodine-biofortified functional fresh products differ from consumers of dietary supplements in their socio-demographics and purchasing motives and therefore need to be strategically positioned and communicated with differently. 

The regression analyses conducted were based on several steps. In the first step, linear and ordinal regression was calculated with all data from the questionnaire on socio-demographic characteristics and buying motives and attitudes. The criteria that were significant in one of the four commodity groups of iodine-containing biofortified foods or in food supplements were noted. There were fourteen of these criteria. In a second step, it was examined how the selection of criteria and the significance levels changed when the four iodine-biofortified fruit and vegetables products were combined into one variable. As the combination did not lead to a significant change in the results of the linear regression analysis, it was decided to combine the iodine-biofortified foods in the presentation in one column and compare them with the food supplements. 

All statistical analyses were carried out using IBM SPSS Statistics 26 (IBM Corp., Armonk, NY, USA). 

## 3. Results

### 3.1. Level of Knowledge of the German Population about the Mineral Micronutrient Iodine

Iodine proved to be one of the best-known trace elements among consumers in Germany. About 84% of the respondents were familiar with iodine in connection with nutrition. In comparison, 89% of respondents were aware of iron and 74% of zinc. Selenium was known by about half of the respondents (54%), while less than one-third was able to associate copper, manganese, cobalt, and molybdenum with nutrition (Appendix A; Figure A1). More than two-thirds of the respondents rated the effect of iodine as “positive”, 28% were “undecided”, and 4% rated the effect as “rather negative” (Appendix A; Table A1). In an open-ended question about how iodine deficiency affects the human body, 55% of respondents indicated a negative effect on the thyroid gland. Another 18% of the participants mentioned fatigue or weakness, respectively, followed by 13% who mentioned the formation of a goiter. In total, the survey participants listed 17 other deficiency symptoms. However, these were mentioned by only 3% or less of the participants. In a subsequent question, respondents were asked to list 10 health impairments resulting from iodine deficiency (Figure 1). Three out of ten symptoms were known by about half or more of the respondents, and these were congruent with the top three of the previous open question. The other seven clinical pictures, such as imbalances of the hormone metabolism and attention deficit hyperactivity syndrome (ADHS), were attributed to iodine deficiency by 4–36% of the surveyed participants. Almost every fourth respondent (24%) stated that they know people with a temporary iodine deficiency from their immediate family environment. Every fifth survey participant was affected by thyroid disease themselves.

Consumers were often unable to assess the contribution of food groups to iodine supply. Half of the survey participants indicated fish and seafood when asked which foods are major sources in their personal iodine supply. Vegetables and fruit were ranked as a very important contributor by 22% and 12% of participants, respectively, while less than 8% attached importance to bread and cereal products and milk and dairy products (Appendix A; Figure A2). Currently, 70% of the respondents use iodized salt to meet their iodine needs (Figure 2). A total of 46% consume fish as a naturally iodine-rich food. A total of 18% consciously buy products made with iodized table salt and labelled accordingly. Only 10% use iodine-containing dietary supplements.

Looking further at different product categories of fresh and processed foods, 58% of respondents were open to products with an increased iodine content to improve their iodine supply. Another 42% were not interested in this type of functional fresh food. Within the group of potential consumers, respondents found meat and meat products most appealing for iodine enrichment. Among foods of plant origin, cereals and cereal products are seen as particularly attractive in this respect, followed by fresh and processed vegetables (Figure 3). Since processed foods can be fortified during artisanal or industrial production, in this study, further consideration focused on fresh fruit, vegetables, and culinary herbs.

### 3.2. Comparison of the Target Groups of Iodine-Biofortified Fruit and Vegetables versus Dietary Supplements

Acceptance of iodine-biofortified foods was dependent on the respondents’ information base and the produce. While additional explanations significantly increased product acceptance for apples and tomatoes, this was not the case for lettuce and basil (Table 2). These varying acceptance levels were related to the frequency of consumption of the produce. While fruits, such as apples, and vegetables are consumed daily by 27% and 17% of Germans, respectively, the figure for lettuce and basil is only 5–6% (Appendix A, Table A2).

Analyses for the purposes of characterizing potential purchasers of iodine-enriched foods show that they differ significantly in several respects from users of dietary supplements (Table 3). Consumers who are attracted to iodine-biofortified fruit and vegetables prefer to buy their groceries for daily needs at farmers’ markets, organic produce shops, or farm stores. Furthermore, this consumer group is interested in foods, such as fish, that are naturally rich in iodine. They do not resort to dietary supplements to improve their supply with essential nutrients. Accordingly, they rate the claim “rich in iodine” for foods as particularly attractive. Likewise, other labels that indicate an increased iodine content in foodstuffs meet with the interest of this group of consumers. When buying food, the target group of iodine-biofortified fruit and vegetables also pays attention to products from integrated cultivation and domestic origin. Typical users of dietary supplements have different characteristics and preferences. They are more often women than men and pay attention to a labelling of functional properties when buying food. In addition, they also rely on dietary supplements for covering their iodine requirement. Interestingly, the users of dietary supplements prefer “biofortified with iodine” as a label for the iodine content, a term that is probably still largely unknown to consumers. Furthermore, they pay attention to ingredients and special health benefits when buying food. This consumer group also likes to use convenience products.

### 3.3. Market Potential for Iodine-Biofortified Fruit and Vegetables in Food Retailing

As shown above, customers of farmers’ markets, organic produce shops, and farm stores are particularly attracted to iodine-biofortified fresh produce. Apple, tomato, and lettuce with this nutritive characteristic are found to be very appealing by four out of ten respondents in this consumer group. The assessment was somewhat weaker for potted basil (Table 4). Consumers who prefer to buy daily groceries in supermarkets and discounters are generally less attracted by the product innovation presented. However, even here, three out of ten respondents said they considered iodine-enriched fruit and vegetables to be very attractive. Once again, acceptance of basil was lower, with only one in five of this customer group very interested in it. Additional explanations of the health benefits of the product by means of an informative text led to an increase in product attractiveness among customers of all shopping locations. On average, however, this effect was less pronounced among customers of supermarkets (8%) than among consumers who buy fruit and vegetables mainly at farmers’ markets, organic produce shops, and farm stores or at discounters (14% and 15%, respectively) (Appendix A, Table A3). In addition to the higher product acceptance of iodine-biofortified foods by customers of farmers’ markets, organic produce shops, and farm stores, it can be noted that 25% of them rate the effect of iodine on the body as clearly positive. This compares to only 15% and 16% of discount and supermarket customers, respectively (Appendix A, Table A4).

Supermarkets are by far the most important place to buy fruit and vegetables in Germany. Slightly more than half of respondents (54%) prefer to buy such fresh produce there. Discounters follow in second place (33%). Farmers’ markets, organic produce shops, and farm stores are the favored shopping locations for 10% of the consumers surveyed. All other sources, including self-supply from one’s own garden, play a minor role (3%). Taking this purchasing behavior into account, supermarkets represent the most important potential distribution channel as regards quantity for iodine-biofortified fruit and vegetables. 

### 3.4. Labelling and Market Launch of Iodine-Biofortified Fruit and Vegetables

In order to make consumers aware of biofortified fruits and vegetables in the purchase situation, labelling of the higher nutrient content is necessary. Overall, the survey participants felt particularly addressed by the labelling “rich in iodine” followed by “high iodine content”. “Biofortified with iodine” received the lowest approval (Figure 4a). Furthermore, several health claims permitted in the European Union can be used in product communication. Three quarters of the respondents rated the claim “contributes to normal thyroid function” as particularly beneficial for their own health. Furthermore, the statements “contributes to a normal production of thyroid hormones”, “contributes to a normal energy metabolism”, and “contributes to a normal function of the nervous system” were rated as valuable for their own health by more than two-thirds of the respondents. The health claim “contributes to normal skin”, on the other hand, received the lowest approval (Figure 4b).

As already shown, the acceptance of iodine-biofortified fruit and vegetables increases when the nutritional and health benefits are communicated to potential customers. However, it was found that this measure did not significantly increase the willingness to pay a higher price for the fresh product presented (Appendix A; Table A5). The analysis of product demand as a function of price showed, independently of the explanatory information, that consumers are willing to pay a price premium of up to € 0.50/kg for apples, based on the usual retail price of € 2.49/kg. With a higher price premium, the willingness to buy dropped significantly. The potential consumer target group was highest at a price increase of 8% per kg of apples. A similar willingness to pay more was found for basil. Here, too, most of the survey participants were willing to pay 8% more for an iodine-rich product. In the case of tomatoes and lettuce, there was an even higher willingness to pay additional prices of 20–25% per kg and unit, respectively (Figure 5).

### 3.5. Consumer Perception of Different Methods to Biofortify Fruit and Vegetables with Iodine

Three options were presented to survey participants as approaches to increasing iodine content in fruit and vegetables. Breeding iodine-rich plant varieties was rated as the most suitable method in five of the six aspects addressed: healthy, trustworthy, effectiveness, naturalness, and environmental friendliness. Only the costs were estimated to be somewhat higher than for the other two techniques, soil fertilization and foliar fertilization during plant cultivation (Figure 6). 

### 3.6. Consumer Perception of Iodine-Biofortified Fruit and Vegetables Compared to Dietary Supplements

In a direct comparison with dietary supplements, iodine-biofortified fruit and vegetables have a number of advantages according to the consumers surveyed. Fresh produce is seen as healthier, more trustworthy, more effective, more natural and environmentally friendly, and less expensive. Only in terms of dosing capability are dietary supplements rated better (Figure 7a). Overall, about 85% of respondents would prefer the iodine-rich foods to improve their iodine supply, while only 15% favor the supplements for this purpose (Figure 7b). Interestingly, even nearly four out of five individuals who use dietary supplements regularly would prefer iodine-biofortified fruits and vegetables to iodine-containing supplements if offered them. Among the group taking supplements daily (*n* = 201 of 1016 respondents), the proportion was 74.1%. Among those using supplements specifically to prevent iodine deficiency (*n* = 99 of 1016 respondents), the proportion was 74.7%.

## 4. Discussion

### 4.1. Germans’ Consumer Knowledge about the Mineral Micronutrient Iodine

The present study shows that iodine is the second-best known mineral micronutrient in the German population. Only iron was better known in this respect. Two-thirds of the respondents were also aware of the positive benefits of iodine on the human organism. This initially suggests that Germans have a high level of knowledge about iodine. However, further consideration of the health-specific effect of iodine reveals some knowledge deficits. About a quarter of the respondents said they did not know how iodine affects the body. Only about half of the respondents were aware of three of the ten positive effects on human health. The results confirm the findings of a recently published German consumer study showing that 57% of the respondents assumed that they knew what the body needs iodine for. A total of 35% did not know this [12]. A comparison with international surveys from Australia and Norway also shows similarities. Between 40% and 62% of the respondents were familiar with iodine-related disease patterns [45,46]. Differences emerge in comparison with developing and emerging countries, such as the Philippines, Nigeria, and South Africa, where more serious knowledge deficits are present [47,48,49]. However, what is common to all the studies mentioned is that foods often cannot be properly assessed in terms of their contribution to iodine supply. In this research, half of the respondents named fish as a major source of their personal iodine intake, followed—with quite a great gap—by vegetables and fruit. Likewise, the above-mentioned Australian study found that almost half of the respondents considered vegetables to be a good source of iodine. Furthermore, one-third of the participants associated fruit as a good source of iodine [45]. In fact, foods of plant origin are generally low in iodine [20]. In Germany, fruit and vegetables account on average for only 3% of the iodine intake of the population. Sea fish also contributes relatively little to iodine supply due to the low quantities consumed. Main sources of iodine intake are milk and dairy products, followed by meat and meat products as well as bread and baked goods [50]. In the present study, only about one in ten of the consumers surveyed considered these food groups to be very important iodine sources. These results are in line with similar findings from Great Britain and Ireland [25,51]. However, it must be critically questioned whether the participants of the studies were always aware of the difference between the iodine content of a food and its contribution to iodine intake. Misunderstandings in this regard would explain, in particular, the high rating of fish. Overall, it remains a challenge to inform consumers better about the sources of iodine in their diets or, alternatively, to find approaches to address consumer expectations regarding the contribution of foodstuffs to iodine intake. In the latter respect, biofortification of fruit and vegetables with iodine seems a suitable approach.

### 4.2. Target Groups of Iodine-Biofortified Fresh Produce versus Dietary Supplements

Slightly more than every second German queried was open to iodine-biofortified foods. In particular, three out of ten respondents were interested in iodine-enriched fresh fruit and vegetables. In a previous study dealing with selenium-biofortified foods, fruit and vegetables were even considered to be the most appealing food group for increasing the content of this micronutrient [29]. It can be concluded from these and other studies that fruit and vegetables biofortified with essential trace elements would meet with relatively high consumer acceptance. In general, consumers are more likely to accept nutritional and health claims if the carrier product itself is considered a particularly healthy food [52,53,54].

Shoppers at farmers’ markets, organic shops, and farm shops felt particularly attracted by iodine-biofortified fruit and vegetables. This consumer group also prefers fish as a natural source for their personal iodine supply as well as food from integrated cultivation and local origin. This suggests that naturalness and sustainability are key purchasing motives for potential buyers of these functional fresh foods. The results are in line with reports from Italy, where consumers stated that biofortified products should be organically grown, as this conveys safety and was associated with a positive effect on health [30]. It was not possible to explain willingness to buy iodine-biofortified fruit and vegetables products by the socio-demographic characteristics age and level of education. This indicates that the product idea appeals to a relatively broad consumer base in this respect. In contrast, many other studies have found that the willingness to try new functional foods decreases with age and increase with education level [54,55,56,57,58]. However, there are also contradictory reports in the literature [29,49]. This inconsistent pattern is probably related to the fact that the surveys were conducted in countries with quite different levels of economic development and dietary habits and, in some cases, also included different nutrients. Therefore, the transferability of these study results to iodine-biofortified foods in Germany is only possible to a limited extent. Overall, the influence of the socio-demographic factors mentioned is likely to depend on the extent to which the foods in question are regularly purchased by the individuals and on the particular nutritional and health claims that are personally considered important. As the relevance of iodine for human health is rather well-known, and fresh fruit and vegetables are consumed regularly by many individuals, iodine-biofortified produce obviously attracts a relatively high level of interest among German consumers. In line with this, it was found that apples and tomatoes, for example, as very popular fresh produce, were found to appeal to significantly more respondents with regard to iodine biofortification than was the case for the less widely consumed potted basil.

Surprisingly, it was found that the potential consumers of iodine-biofortified fruit and vegetables do not pay attention to the ingredients of their food when buying it. A possible reason for this observation could be that the target group has a high level of trust in the supplier and product based on the place of purchase and therefore pays less attention to specific ingredients. In this respect, there is obviously a difference to other functional foods, which so far have been mainly processed products. Here, users pay more attention to the declared ingredients and health benefits [59,60,61]. Likewise, this applies to users of dietary supplements, as shown by results of the present study and reports in the literature [62,63]. Women are more highly represented among dietary supplement users than men. In addition to supplements and processed foods labeled with health effects, this consumer group also purchases convenience products more frequently. In summary, it is evident that potential customers for iodine-biofortified functional fresh foods and users of dietary supplements are quite different types of consumers. Users of dietary supplements are strongly focused on ingredients and health benefits, possibly also to compensate for deficits resulting from a sometimes less healthy diet resulting from the consumption of convenience food [64]. In contrast, the target group for iodine-biofortified products values fresh food and food with little processing that has been produced in a natural and sustainable way. However, even about three-quarters of those who currently consume dietary supplements daily would prefer iodine-biofortified products in a direct comparison.

### 4.3. Market Potential for Iodine-Biofortified Fruit and Vegetabeles in Different Food-Trade Channels

Even if customers of farmers’ markets, organic produce shops, and farm stores felt particularly attracted by iodine-biofortified fruit and vegetables, only a limited number of buyers can be reached via these distribution channels. Overall, only every tenth respondent prefers these shopping locations for purchasing daily groceries. However, marketing via this sales channel has the advantage that there is a direct exchange between customers and retailers. This plays an important role for products that require special explanation, such as biofortified produce. On the other hand, this distribution route is also associated with disadvantages. Fresh foods are usually offered here as loose goods. This results in a higher risk of them being mixed-up with visually indistinguishable common goods. In particular, this applies to foodstuffs, such as apples or tomatoes, of which various varieties are displayed. The labeling of basic nutritional values, as is required by law in the European Union when individual ingredients such as iodine are advertised [31], also presents a difficulty in the case of unpackaged goods. Supermarkets are important outlets for food purchasing in Germany. Slightly more than half of the respondents usually buy their fruit and vegetables here. About three out of ten supermarket shoppers rated iodine-biofortified fruit and vegetables as very appealing. Supermarket customers with a strong interest in iodine-rich fruit and vegetables represent a total of 15% of the consumers surveyed. This suggests relatively large market potential for product innovation in this area of food retailing. However, to tap this potential, an appropriate market launch strategy must be in place.

### 4.4. Conceptual Market Launch of Iodine-Biofortified Foods

To highlight the special nutritional value of iodine-biofortified fruit and vegetables, various claims were questioned. Here, two-thirds of the respondents rated the claim “rich in iodine” as particularly appealing. Similar observations were made in a previous study on selenium-biofortified apples [26]. The claim “biofortified with iodine” was not convincing in the survey. Less than one in three respondents rated the claim as appealing. The reason for this may be the unfamiliarity of German consumers with the concept and term of biofortification. Here, there is a similarity with reports from other European countries [30,65]. With regard to health claims, statements relating to the thyroid gland were particularly well received. Three quarters of respondents would, for example, rate the claim that “iodine contributes to normal thyroid function” as appealing when buying fruit and vegetables. This is consistent with the finding reported above that about two-thirds of respondents can classify iodine deficiency as a cause of thyroid disorders. Almost one in five of the study participants stated that they themselves are affected by thyroid disease. Epidemiological studies indicate that, in Germany, one in three adults has a pathological change in the thyroid gland, but many of those affected are still unaware of this [7,66]. Individuals affected by thyroid disease were slightly more attracted to iodine-biofortified fruits and vegetables than the overall sample. Even more pronounced effects were found in relation to the diet. Four out of ten vegans and vegetarians felt strongly positive about the concept of iodine-biofortified fruits and vegetables. Among flexitarians and individuals with no special dietary habits, this proportion was lower.

Health claims other than those related to thyroid function that are permitted for iodine-enriched foods according to the EU Health Claims Regulation [1] also still met with a relatively high level of consumer acceptance. For example, more than every second respondent said that statements about the importance of iodine in energy metabolism or in maintaining normal skin were considered useful for their own health. The latter information was of particular interest to younger adults aged 18–34. Likewise, Krystallis et al. observed an age-dependent attractiveness of various health-related statements. Young people were particularly interested in functional foods that improve physical fitness and energy levels, while older people were focused on avoiding health risks [67].

The use of nutrition and health claims in the advertising of iodine-biofortified foods requires that the iodine content in the products reaches at least 22.5 µg per 100 g of fresh weight. This level can be achieved, for example, by using soil or foliar iodine fertilization in the field production of horticultural crops [26,68,69]. Another option is provided by hydroponic systems in greenhouses, where plants are cultivated in an iodine-containing nutrient solution [24,25,70]. In general, fertilizer-based biofortification approaches tend to be viewed with some skepticism by consumers, as has been previously reported [29]. However, breeding methods are not suitable for increasing the iodine content in food crops because the uptake of this trace element by plants is basically limited by its low content in the soil [71,72]. Many consumers also take a rather critical view of plant production in soilless systems although such methods are already very widespread today, for example, for the cultivation of fruit vegetables, such as tomatoes or cucumbers [73,74,75]. Hence, for the marketing of biofortified food, addressing the enrichment pathway is less likely to generate consumer interest.

Iodine-biofortified fruit and vegetables are not yet widely available in Europe. In Italy, however, they were successfully introduced to the market some years ago and are increasingly in demand there [30]. Further promotion of such functional fresh foods requires that all actors in the food-value chain can generate their own benefit from this. In that respect, it is important to know to what extent consumers are willing to pay a higher price for biofortified fresh produce. In a study of selenium- and iodine-biofortified apples, Kleine-Kalmer et al. found that popular apple varieties and an appropriate price premium are decisive for the purchase decision [76]. A meta-analysis on this matter revealed that consumers are, on average willing to pay 21% more for biofortified foods [77]. Values at a similar or slightly higher level were determined in this study for apples, tomatoes, and lettuces. Only for potted basil, a product consumed less frequently, was willingness to pay a higher price lower, at a maximum of 8%.

To stimulate demand for biofortified foods, several accompanying measures can be useful. In Germany, for example, there is the Arbeitskreis Jodmangel e.V. (Working Group on Iodine Deficiency), scientifically guided by medics and nutritionists, which regularly informs via media releases, brochures, and events on the importance of iodine in the diet and the current iodine-supply status of the population. Studies from other countries also show that broad-based information campaigns about the added value of biofortified products and their contribution to human health are helpful in increasing the acceptance of biofortified foods [78,79,80].

As mentioned above, supermarkets are the most important distribution channel for fruit and vegetables in Germany. For a successful market launch of biofortified foods in these shopping outlets, an appropriate presentation of the goods at the point of sale is crucial [29]. For this purpose, a conspicuous special placement of the product could be helpful, accompanied by large-format posters on which the essential nutritional and health benefits of the products are presented in a concise form. Furthermore, suitable nutrition and health claims (“rich in iodine”, “iodine contributes to normal thyroid function”) should be placed on the packaging of the products. Due to the limited space available here, customers can be referred to an established product website for more detailed information, for example, via a QR code. With the marketing concept outlined, selenium-enriched apples were recently successfully launched as the first biofortified fresh produce in German food retailing [81].

### 4.5. Limitations of the Present Study and Implications for Further Research

In the present study, consumers were analyzed with regard to their attitudes towards a functional fresh food category that is still largely unknown in this form in Germany. Concrete examples were presented to the respondents so that they could better imagine the product innovations. However, it was possible to include only a small selection of fresh foods as examples. The results show that acceptance and willingness to pay additional prices are strongly dependent on the product in question. In order to clarify more precisely how specific health claims, packaging designs, sales prices, or other factors affect the purchase probability for iodine-biofortified fruit and vegetables, more in-depth, methodological approaches, such as conjoint analysis, should be included in follow-up studies. However, even in this way, the actual purchase situation can only be simulated approximately. Therefore, real-life sales experiments in food retailing would be useful in the course of preparing a market launch. 

## 5. Conclusions

The introduction of iodine-biofortified fruit and vegetables into the food trade opens up a novel approach to improving the iodine supply of the population, which can usefully complement existing prophylactic measures, such as iodization of table salt. The present consumer survey has shown that six out of ten Germans are open to the product idea, and even 85% of consumers would prefer an iodine-biofortified fruit or vegetable to a food supplement. These high approval ratings are probably due to the fact that iodine is a relatively well-known trace element. Many consumers also have an idea about its health benefits, even if this is mostly limited to a few aspects. As a potential target group for this functional fresh food, people who attach particular importance to natural and sustainable food production were identified. These individuals prefer to buy everyday groceries at farmers’ markets, organic produce shops, or farm stores. However, this distribution channel has a limited buyer base due to its relatively low importance in the food trade. The largest market potential for iodine-biofortified produce is offered by supermarkets. When launching fruit and vegetables rich in iodine, it is important to highlight the special nutritional and health benefits at the point of sale. In addition, a convincing packaging design with labeling of attractive health claims and appropriate pricing of the product are important. The market entry will ideally be accompanied by a broad information campaign by scientifically based institutions from the fields of medicine and nutritional sciences. A successful realization of this concept could contribute to a sustainable improvement of the health status of the population in Germany and other countries affected by iodine deficiency.

## Figures and Tables

**Figure 1 nutrients-13-04198-f001:**
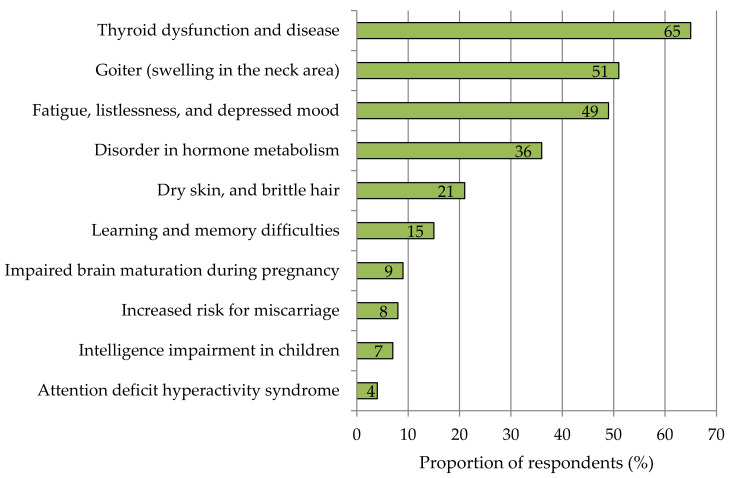
Respondents’ estimation of which of the health impairments presented can be attributed to iodine deficiency. Multiple answers were allowed. *n* = 1016.

**Figure 2 nutrients-13-04198-f002:**
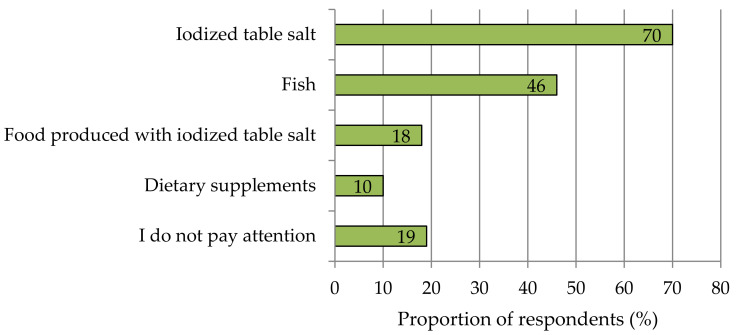
Products currently used to prevent possible iodine deficiency. Multiple answers were allowed. *n* = 1016.

**Figure 3 nutrients-13-04198-f003:**
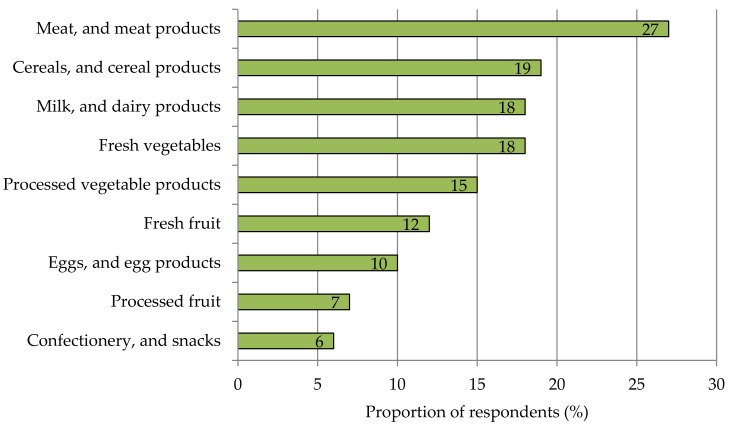
Acceptance of different food categories to optimize iodine intake through the daily diet. Multiple answers were allowed. *n* = 1016.

**Figure 4 nutrients-13-04198-f004:**
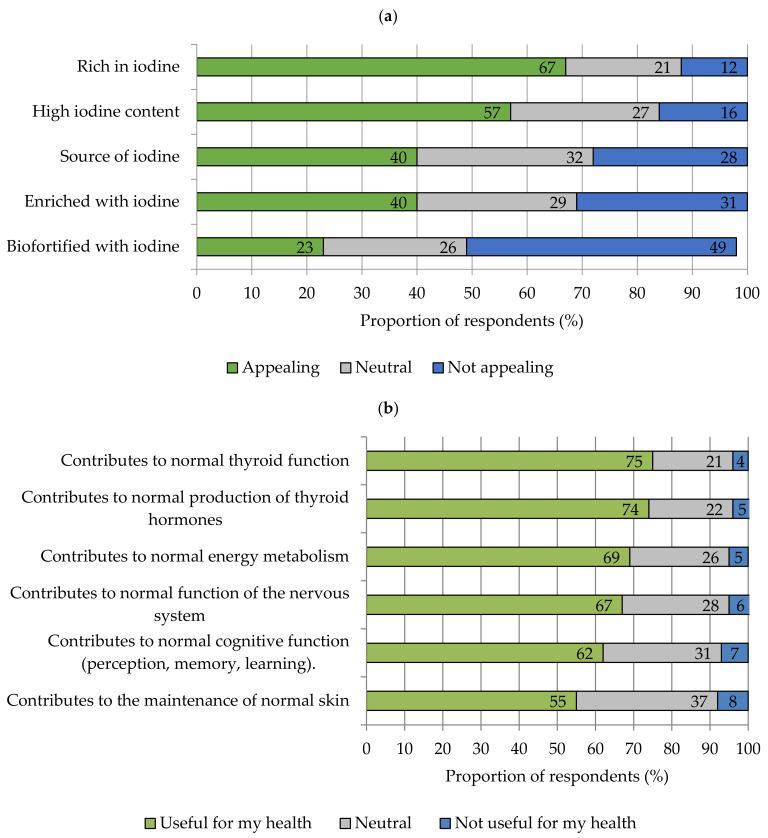
Acceptance of nutrition and health claims for iodine-rich fresh fruit and vegetables study. Participants were asked: (**a**) please rate the following labelling for iodine-biofortified fresh fruit and vegetables. *n* = 1016, and (**b**) please assess whether the following effects of iodine-biofortified fresh fruit and vegetables are personally useful to you. *n* = 508.

**Figure 5 nutrients-13-04198-f005:**
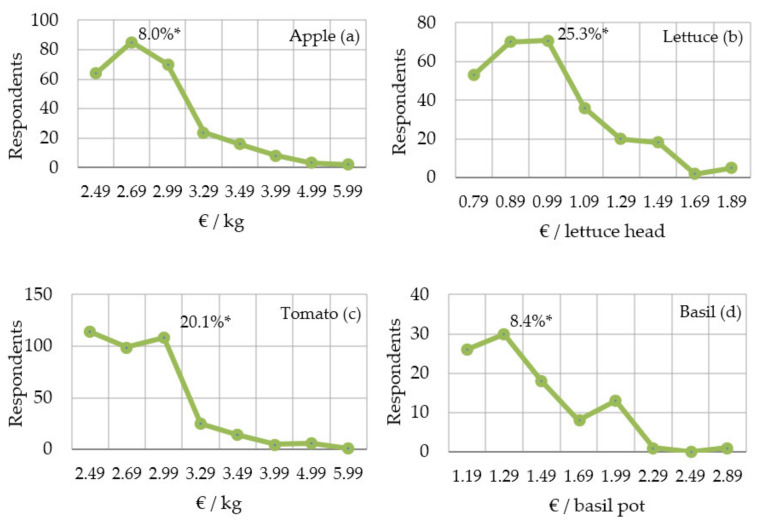
Development of willingness to pay for iodine-biofortified apples (**a**), lettuce (**b**), tomato (**c**), and potted basil (**d**) based on prices per kilogram. The lowest price always represents the current average market price. *n* = 1016. * Relative willingness to pay more with the highest number of potential customers.

**Figure 6 nutrients-13-04198-f006:**
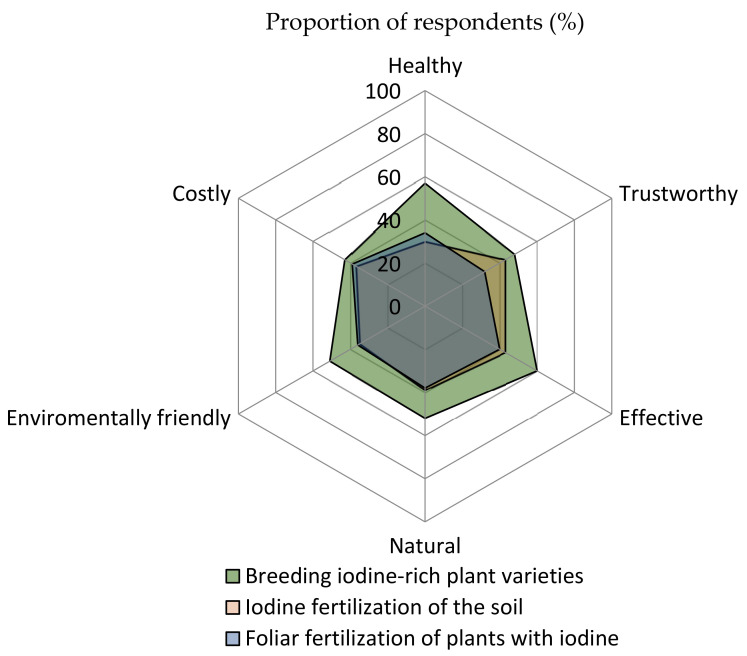
Consumer evaluation of different approaches to increasing iodine content in fruit and vegetables. *n* = 500–514.

**Figure 7 nutrients-13-04198-f007:**
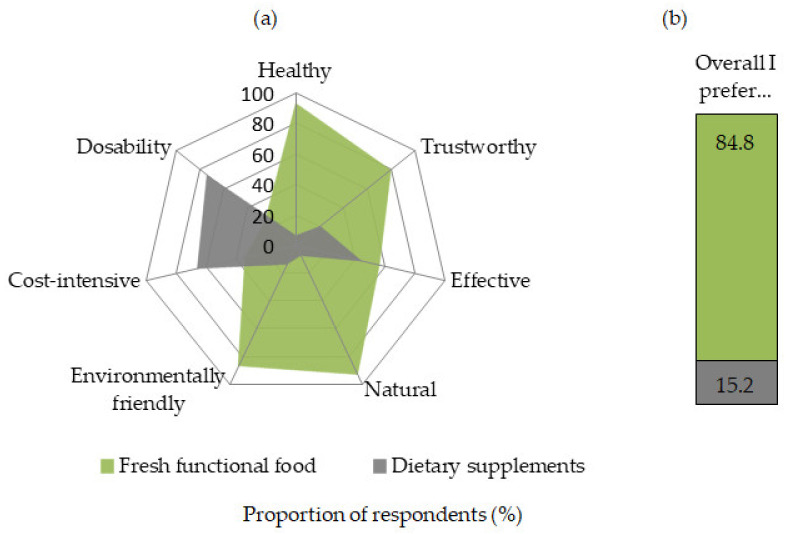
Consumer evaluation of iodine-containing fruit and vegetables and dietary supplements based on various decision factors (**a**) and a comparison of preferences as a whole (**b**). *n* = 1016.

**Table 1 nutrients-13-04198-t001:** Attributes of the consumer sample (specified criteria for approximate representativeness).

Characteristics		Sample (in %)
Consumption of fresh fruit, and vegetables	Consume at least rarely	100
Gender	Male	50
	Female	50
	18–24	10
	25–34	18
Age	35–44	17
	45–54	23
	>55	32
Region in Germany	North	18
	West	36
	South	28
	East	18
Responsible for purchasing	Mainly myself	64
	Myself and another person	36

**Table 2 nutrients-13-04198-t002:** Effect of explanatory information on consumer acceptance of iodine-biofortified fresh fruit and vegetables. *n* = 1016.

Product	With Information Text (Mean Value)	Without Information Text (Mean Value)	Test of Difference ^1^
Apple	0.76	0.60	F(1.1014) = 3.939, *p* = 0.047 *
Tomato	0.86	0.62	F(1.1014) = 8.463, *p* = 0.004 **
Lettuce	0.73	0.62	F(1.1014) = 2.044, *p* = 0.153
Basil	0.42	0.37	F(1.1014) = 0.465, *p* = 0.495

^1^ F(dfCounter, dfDenominator), F-Value; *p*, significance; ** high significance; * significant.

**Table 3 nutrients-13-04198-t003:** Results of the regression analysis of concept acceptance. *n* = 1016.

	Iodine-Biofortified Food R^2^ = 0.395	Supplements Nagelkerke R^2^ = 0.210
Sociodemographic variable		
Female Gender	−0.033 (−1.296)	0.406 ** (9.790)
Shopping location		
Farmers’ market, organic produce shop, or farm store	0.056 * (2.226)	−0.254 (1.499)
Attention to labelling		
Attention to functional food labelling when buying food	0.048 (1.769)	0.377 ** (22.388)
Personal iodine sources (currently)		
Fish as a source of iodine	0.052 * (2.018)	0.086 (0.427)
Dietary supplements as a source of iodine	0.043 (1.683)	1.946 ** (35.787)
Labelling iodine content		
High iodine content	0.146 ** (4.026)	0.011 (0.020)
Enriched with iodine	0.134 ** (4.317)	0.075 (1.335)
Iodine-biofortified	0.118 ** (4.003)	0.184 ** (9.317)
Rich in iodine	0.272 ** (7.891)	0.021 (0.069)
Purchase motives		
Integrated cultivation	0.097 ** (3.553)	0.028 (0.192)
Produced in Germany	0.087 ** (3.190)	−0.069 (0.713)
Focus on food ingredients	−0.054 (−1.933)	0.187 ** (8.353)
Never buying products with specific health effects	−0.039 (−1.532)	−0.151 * (6.039)
Often buying ready-made dishes	−0.031 (−1.191)	0.218 ** (13.531)

The table shows the results of the linear and ordinal regression analyses. Linear regression was used for the iodine biofortified foods. The first value per column indicates the standardized beta coefficient; the asterisks symbolize the significance level. ** = α < 0.01 highly significant; * = α < 0.05 significant. The value in brackets stands for the *t*-value. Ordinal regression was used for the supplements. The first value per column indicates the estimator; the asterisks symbolize the significance level. ** = α < 0.01 highly significant; * = α < 0.05 significant. The value in brackets stands for the Wald value. Nagelkerke R^2^ = 0.210; Pearson´s Chi Quadrat 2042.070, significant 0.261.

**Table 4 nutrients-13-04198-t004:** Proportion of respondents rating the iodine-biofortified fresh products presented as very appealing as affected by the location of purchase. (*n* = 1016).

Product	Customers of Farmers’ Markets, Organic Produce Shops, and Farm Stores	Discount Consumers	Supermarket Consumers
Apple	41.8%	32.6%	29.7%
Tomato	41.8%	36.8%	34.7%
Lettuce	42.9%	27.7%	28.2%
Basil	26.8%	19.0%	19.8%

Mean value based on acceptance question for biofortified fruits and vegetables (Appendix B Q18 and Q19).

## Data Availability

The data presented in this study are available on request from the corresponding author. The data will be made publicly to a later stage.

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
