# Peer review of "Consumer Acceptance and Market Potential of Iodine-Biofortified Fruit and Vegetables in Germany"

_nutrients, 2021, doi:10.3390/nu13124198_

Round 1
Reviewer 1 Report
The main aim of paper: ‘Consumer Acceptance and Market Potential of Iodine-Biofortified Fruit and Vegetables in Germany’ was to examine the acceptance of German consumers towards iodine-biofortified fruit and vegetables, to characterize the target groups and to estimate the market potential in food retailing. From my point of view, the topic of the study is important and really interesting.
In general, the procedure, the study design and statistical analyses are well organized.
Moreover, the topic is quite original and the obtained results in general confirmed the concept indicating that when you introduce the ‘new’ food product into the market (e.g. you launch fruit and vegetables rich in iodine) you have to underline the issue referring to the special nutritional and health benefits.
However, I would like to indicate some major suggestions for Authors, because in my opinion the Authors should reorganized the final version of manuscript.
These are some main suggestions for Authors:
In the Discussion part you do not have to include information referring to the particular tables/figures; just concentrate on the most important results and info from the resources. You also do not include info regarding the percentage (because you repeat again the same info that was mentioned in the Result section).
In the Conclusion section focus strictly on the main conclusions (2-3 main conclusions). So, please reorganize this part of the paper; some paragraphs are not the direct conclusions actually.
Are there any additional practical implications of your findings?
Author Response
Dear Reviewer,
Please find attached our point-by-point response to your comments.
We thank you very much for your suggestions and hope that we have implemented them according to your ideas.
Kind regards
Ann-Kristin Welk

Reviewer 2 Report
I think this paper can be an important basis for consumer researches based on the biofortification. As we do not know much about consumer preferences on this topic, yet. What I miss from the paper is a deeper methodological view. However, this may not be necessary, as until now not many studies are available with this topic.
Major concerns:
- All data in this research comes from personal statements of respondents. Researchers are aware of this, and they mention this in the Limitations chapter. I strongly think that this should be mentioned also in other chapters of the paper. E.g. respondents say that they would buy it, they say that they would pay more, they say that they are aware of health effects of iodine etc. These are far from the fact, these are just personal statements for now.
- I would like to add the following to the foregoing: authors state in the line 564 that “the claim "Rich in Iodine" proved to be particularly effective”. Authors have only descriptive results from respondents. Which is a very important collection of results. However, I strongly argue against stating that anything is “proved” here yet regarding to claims. There are many important papers that try to prove that claims do not help selling or help selling functional foods. We should not rely 100% of what respondents say.
- I think that sometimes ethical statement is not needed for surveys that collect anonym and not vulnerable data. However, I was surprised that no ethical statement is available for this research which collected data about respondents’ diseases. Please, let us know why you consider it unnecessary (e.g. anonym data, not vulnerable data, etc.)
Minor comment:
- Authors write in the line 136 that “interviews that did not pass the question on the level of attention were excluded from the sample”. Please add the information how you measured attention.
- Age groups in Table 1 has an option “45-44”. Please correct it.
- In line 323 you write “Chi Quadrat Person”. Do you mean “Pearson’s chi-squared”?
- Please change the title of Table 4. Readers do not want to backlink to the appendix to understand what were the “question 18 and question 19”.
Author Response

(The authors gave the same response as above.)

Reviewer 3 Report
The subject matter is of great scientific interest, and reading the article it is clear that the authors have a profound knowledge of the issues involved.
Granted that the article is well structured, has an appropriate research design, I would like to stress that for the level of detail and the breadth of the questionnaire used, of great interest, the many topics covered could very well be used to make two separate papers in order to give more space to the various aspects. In fact, the paper appears to be too long and divided into too many sub-sections. It might be useful to make it a little more streamlined, perhaps by trying to merge some sub-sections, such as 3.5.
The title summarises the essential information of the paper, with a bias towards economic issues (marketing, consumer behaviour analysis, etc.) which also emerges in the well-done abstract, which offers a brief but clear description of the paper and its objectives, and in the research questions.
The introduction, also well articulated, lets the reader know what the research questions are, its level of importance and context. It would be useful, although already quite extensive, to expand the literature by introducing some bibliographic references relating to the factors influencing consumer acceptance of biofortified products (some examples):
Rizwan, M., Zhu, Y., Qing, P., Zhang, D., Ahmed, U. I., Xu, H., ... & Tariq, A. (2021). Factors Determining Consumer Acceptance of Biofortified Food: Case of Zinc-Fortified Wheat in Pakistan's Punjab Province. Frontiers in Nutrition, 8.
Murekezi, A., Oparinde, A., & Birol, E. (2017). Consumer market segments for biofortified iron beans in Rwanda: Evidence from a hedonic testing study. Food Policy, 66, 35-49;
Birol, E., Meenakshi, J. V., Oparinde, A., Perez, S., & Tomlins, K. (2015). Developing country consumers' acceptance of biofortified foods: a synthesis. Food Security, 7(3), 555-568.
Etc.
In the paragraph on materials and methods
Line 132: Explain what is meant by "Respondi AG".
Line 134: Explain what the software used is (LimeSurvey )
Line 160 Explain what is meant by "concept test".
Questions 11 and 12 of the questionnaire assume a technical knowledge of the subject matter. It seems difficult for any respondent to have a clear opinion on the effects of iodine deficiency on human health. Such specific questions perhaps presuppose the use of specific samples, perhaps the method of choosing the interview sample should be better explained.
The results are in line with the methodology and are very comprehensive.
The discussion paragraph seems excessively extensive and should be reduced a little
Figures and tables are appropriate, show the data correctly and are easy to interpret and understand.
Table 1: The sum of the percentages of the Region in German is 97 a not 100.
Author Response

(The authors gave the same response as above.)

Round 2
Reviewer 1 Report
Dear Authors,
Thank you for applying my all suggestions.
Kind regards,